# The ATP Synthase Subunits *FfATPh*, *FfATP5*, and *FfATPb* Regulate the Development, Pathogenicity, and Fungicide Sensitivity of *Fusarium fujikuroi*

**DOI:** 10.3390/ijms241713273

**Published:** 2023-08-26

**Authors:** Xin Yang, Zhili Yuan, Xiaowei Cai, Shuai Gui, Mingguo Zhou, Yiping Hou

**Affiliations:** College of Plant Protection, Nanjing Agricultural University, Nanjing 210095, China; 2020102118@stu.njau.edu.cn (X.Y.); 2023202075@stu.njau.edu.cn (Z.Y.); 2021102113@stu.njau.edu.cn (X.C.); 15550843363@163.com (S.G.); mgzhou@njau.edu.cn (M.Z.)

**Keywords:** *Fusarium fujikuroi*, ATP synthase, *FfATPh*, *FfATP5*, *FfATPb*, pathogenicity

## Abstract

ATP synthase catalyzes the synthesis of ATP by consuming the proton electrochemical gradient, which is essential for maintaining the life activity of organisms. The peripheral stalk belongs to ATP synthase and plays an important supporting role in the structure of ATP synthase, but their regulation in filamentous fungi are not yet known. Here, we characterized the subunits of the peripheral stalk, *FfATPh*, *FfATP5*, and *FfATPb*, and explored their functions on development and pathogenicity of *Fusarium Fujikuroi*. The *FfATPh*, *FfATP5*, and *FfATPb* deletion mutations (*∆FfATPh*, *∆FfATP5*, and *∆FfATPb*) presented deficiencies in vegetative growth, sporulation, and pathogenicity. The sensitivity of *∆FfATPh*, *∆FfATP5*, and *∆FfATPb* to fludioxonil, phenamacril, pyraclostrobine, and fluazinam decreased. In addition, *∆FfATPh* exhibited decreased sensitivity to ionic stress and osmotic stress, and *∆FfATPb* and *∆FfATP5* were more sensitive to oxidative stress. *FfATPh*, *FfATP5*, and *FfATPb* were located on the mitochondria, and *∆FfATPh*, *∆FfATPb*, and *∆FfATP5* disrupted mitochondrial location. Furthermore, we demonstrated the interaction among FfATPh, FfATP5, and FfATPb by Bimolecular Fluorescent Complimentary (BiFC) analysis. In conclusion, *FfATPh*, *FfATP5*, and *FfATPb* participated in regulating development, pathogenicity, and sensitivity to fungicides and stress factors in *F. fujikuroi*.

## 1. Introduction

Rice Bakanae Disease (RBD) is a worldwide fungal disease, which is a systemic disease affecting the above-ground part of rice [1,2]. *Fusarium fujikuroi* is the main pathogen of RBD [3], it is a destructive filamentous fungus that can infect rice, soybeans, corn, and many other crops, causing crop reduction [4,5]. Studies have found that *F. fujikuroi* can produce gibberellins (GAs) during the infection process of the host. Although GA is a plant hormone, it is considered a secondary metabolite (SM) of *F. fujikuroi* and enhances the pathogen’s virulence [6]. In addition, other secondary metabolites, such as fumonitoxin (FBs), monocybin (MON), and Beauveria bassiana toxin (BEA), are also produced during the infection of *F. fujikuroi* [7,8,9], which can affect the intracellular hormone balance, resulting in cytotoxicity and eventually lead to crop death. At the same time, these toxic metabolites are also threatening the health of humans and animals [10].

ATP is essential for all cellular functions in a living organism [11]. F_1_F_0_-ATP synthase is located at the end of the mitochondrial respiratory chain and synthesizes most of the ATP during oxidative phosphorylation [12,13]. In *Fusarium oxysporum*, knock-out mutants of ATP synthase genes resulted in relatively weak virulence compared to the wild-type [14]. In *Candida albicans*, the *Δatp2* mutant exhibited an immediate and sharp reduction in cell viability on non-fermentable carbon sources; corresponding to this, the *Δatp2* mutant displayed avirulence in a murine model of disseminated candidiasis [15]. In *Ustilago maydis*, deletion of the *ATP20* induces ROS stress [16]. Altogether, ATP synthase plays a significant role in vegetative growth and the pathogenicity of phytopathogenic fungi.

The F_1_F_0_-ATP synthase consists of three parts: the hydrophilic F_1_ head, the lipophilic F_0_ base, and the peripheral stalk [17,18], and the assembly of the peripheral stalk is the critical step for assembling a functional ATP synthase [19]. The peripheral stalk of mammals includes subunits b, d, F6, and 5 [20]. In *Saccharomyces cerevisiae*, ATPb has an N-terminal hydrophobic domain and a C-terminal hydrophilic domain, which can explain its association with F_1_ and F_0_ [21]. The stator subunit ATP5 can prevent the normal connection of F_0_ to F_1_ in the inner membrane vesicles where ATP5 is absent, which further proves that the stator is vital for the stability of the subunit 9 ring and F_1_ complex [22]. Previous studies on the topological structure of the ATPh in the complex revealed its central position in the peripheral stalk, which may play an essential role in the physical coupling between F_1_ and F_0_ and the stability of the peripheral stalk [23]. Studies have shown that the peripheral stalk is of great significance to Eukaryotes. However, the regulation of the peripheral stalk in filamentous fungi remains unclear currently.

In this experiment, we identified the *FfATPh*, *FfATP5*, and *FfATPb* in *F. fujikuroi* and confirmed that these three genes play a vital role in vegetative growth, pathogenicity, stress responses, and sensitivity to fungicides. Our research provides guidance for the comprehensive control of RBD.

## 2. Results

### 2.1. Identification of FfATPh, FfATP5, and FfATPb

The homologous genes of *ATPh*, *ATPb*, and *ATP5* in *Saccharomyces cerevisiae* were found by BLAST in the *Fusarium* genome database as *FfATPh*, *FfATP5*, and *FfATPb*. *FfATPh* is 729 bp in length, contains two predicted introns, and it is predicted to encode a 128 amino-acid protein, which contains one domain of ATP_sub_h. *FfATPb*, a 934 bp gene with three introns, is predicted to encode a 242 amino acid protein. It consists of a long continuous curved α-helix extending from the top of the F_1_ domain to the surface of the membrane domain, with an N-terminal hydrophobic domain and a C-terminal hydrophilic domain. *FfATP5* is 1045 bp in length with three predicted introns, encodes 226 amino acids, and contains one domain of OSCP, an oligomycin sensitivity-conferring protein (Figure 1b). Phylogenic analysis showed they were homologous to *Fusarium oxysporum* and *Fusarium verticillioides* but not as much to *Saccharomyces cerevisiae*, *Botrytis cinerea*, or *Aspergillus nidulans* (Figure 1a).

### 2.2. Regulation of FfATPh, FfATP5, and FfATPb in Vegetative Growth

We inoculated each strain on PDA, V8, CM, and MM media to examine the function of *FfATPh*, *FfATP5*, and *FfATPb*. Compared to the wild-type (WT), the mycelial growth of the mutants was slower, aerial hyphae were reduced, and there were obvious morphological defects in *F. fujikuroi* (Figure 2a). In yeast, the deletion of *ATPh* led to the inability to utilize nonfermentable carbon sources [23]. An observation of the mycelial morphology showed that the mycelia of *ΔFfATPh*, *ΔFfATP5*, and *ΔFfATPb* had more branches and grew more densely (Figure 2b). Furthermore, the gene-deletion mutants lost sporulation ability (Figure 2d), which will affect the diffusion ability of *F. fujikuroi* in the field.

### 2.3. FfATPh, FfATP5, and FfATPb Regulate the Sensitivity to Different Fungicides

The sensitivities of *FfATPh*, *FfATP5*, and *FfATPb* deletion mutants to 0.125 μg/mL tebuconazole, 0.0625 μg/mL fludioxonil, 0.1 μg/mL prochloraz, 0.25 μg/mL phenamacril, 5 μg/mL pyraclostrobine, and 0.25 μg/mL fluazinam were determined (Figure 3). Compared with the WT, the sensitivity of *FfATPh*, *FfATP5*, and *FfATPb* deletion mutants to fludioxonil, phenamacril, pyraclostrobine, and fluazinam decreased, while the sensitivity to prochloraz did not change significantly (Figure 3b). In addition, the susceptibility of *∆FfATPh* and *∆FfATP5* to tebuconazole and *∆FfATPb* to fludioxonil declined, while the sensitivity of *∆FfATP5* to fludioxonil increased. It is suggested that *FfATPh*, *FfATP5*, and *FfATPb* nonspecifically regulated the sensitivity of *F. fujikuroi* to different fungicides.

### 2.4. FfATPh, FfATP5, and FfATPb Are Involved in the Regulation of the Sensitivity to Various Stresses

To investigate how *FfATPh*, *FfATP5*, and *FfATPb* respond to environmental stresses, we set up five types of stress: ionic stress (Na^+^, K^+^, Ca^2+^, Mg^2+^, and Zn^2+^), osmotic stress (Sor), a cell wall-damaging agent (Congo Red), a cell membrane-damaging agent (SDS), and an oxidant agent (Menadione). The results showed that the sensitivity of *ΔFfATPh-5 and ΔFfATPh-7* to ionic stress (Na^+^, K^+^, Ca^2+^, Mg^2+^, and Zn^2+^) and osmotic stress decreased significantly (Figure 4). *ΔFfATPb* and *ΔFfATP5* exhibited more sensitivity to oxidative stress (Figure 5). Overall, although all three genes belong to the peripheral stalk of ATP synthase, they may also be involved in different metabolic pathways in fungi, resulting in different responses to stress.

### 2.5. FfATPh, FfATPb, and FfATP5 Are Indispensable for Pathogenicity

*F. fujikuroi* produces a variety of secondary metabolites (SMs), the most important of which is gibberellin (GA), a hormone that causes rice plants to grow exceptionally long [6]. To investigate the function of *FfATPh*, *FfATP5*, and *FfATPb* in *F. fujikuroi*, the wild-type and deletion mutants were inoculated on rice seedlings for 7 days, and the result showed that the length of the rice shoot sheaths was significantly higher than that of the deletion mutants (Figure 6). Combined with the information from previous studies, the mutant strains were unable to produce conidia. Therefore, *FfATPh*, *FfATP5*, and *FfATPb* are essential for pathogenicity in *F. fujikuroi*.

### 2.6. FfATPh, FfATP5, and FfATPb Are Located in Mitochondria

To determine the location of FfATPh, FfATP5, and FfATPb, we added a GFP tag to the C-terminal of each gene and observed the subcellular localization of the gene with a TCS SP8 confocal microscope (Leica TCS SP8, Germany). At the time, the mycelia were stained with Mitro-Tracker (Mitochondrial dye), and we observed the co-localization between each gene and mitochondria. Based on the experimental results, we concluded that FfATPh, FfATP5, and FfATPb are located in mitochondria (Figure 7).

### 2.7. FfATPh, FfATP5, and FfATPb Regulate the Localization of Mitochondria

Previous studies have found a connection between ATP synthase and the structure of mitochondria, where the oligomerization of ATP synthase is involved in the formation of mitochondrial cristae [24]. In *S. cerevisiae*, the absence of *ATPb* can lead to the appearance of onion-like structures within mitochondria [25]. The deletion of *ATPh* can also severely influence the assembly of ATP synthase [26], causing it to scatter alone on the cell membrane. Therefore, we speculate that the absence of *FfATPh*, *FfATP5*, and *FfATPb* will also affect the structure of mitochondria. Using Mito-Tracker, the hyphae of wild-type A and deletion mutants were stained, and it was found that mitochondria in the wild-type strain showed a filamentous distribution. In the mutant strains, mitochondria were uniformly scattered in the cytoplasm (Figure 8), which indicated that the deletion of the three genes severely affects the distribution of mitochondria.

### 2.8. Interaction among FfATPh, FfATP5, and FfATPb

In *S. cerevisiae*, the peripheral stalk is an essential component in offsetting the torque generated from proton translocation during ATP synthesis or ATP synthesis hydrolysis during proton pumping [27]. Previous studies have shown that in addition to interacting with some subunits on the F_1_ and F_0_ domains, there were also interactions between the several subunits that make up the peripheral stalk [28,29]. This study explored the interactions among FfATPb, FfATP5, and FfATPh using Co-IP, BIFC, and yeast two-hybrid assays. The result of BIFC demonstrated an interactive relationship between FfATPb and FfATP5, FfATPh and FfATP5, and FfATPb and FfATPh (Figure 9), and the interaction between FfATPb and FfATP5 and FfATPb and FfATPh was further confirmed through Co-IP assays (Figure 10), while yeast two-hybrid assays indicated that there was no interaction among the three proteins (Appendix A).

## 3. Discussion

ATP is the common energy currency of cells, and F_1_F_0_-ATP synthetase is the site that catalyzes the synthesis of ATP, so the normal function of ATP synthetase is the premise of maintaining the operation of organisms [30,31,32]. The peripheral stalk is a crucial component of ATP synthase; it can prevent the α- and β-subunits of F_1_ from co-rotating with F_0_ [33], and the loss of any subunit can lead to the paralysis of ATP synthase. In this study, we identified the subunits of the peripheral stalk, *FfATPh*, *FfATP5*, and *FfATPb*, in *F. fujikuroi* and explored their biological functions.

In *S. cerevisiae*, the lack of *ATPh* severely affects the growth of yeast cellular on nonfermentable sources, which is attributed to defects in the assembly/stability of the ATP synthase [34]. In Arabidopsis, reduced expression of the *ATP5* gene leads to the stunting of dark-grown seedlings and the downward curling or wavy-edged leaf margins of light-grown plants. The antisense expression of *ATP5* reduced the total ATP level of dark-grown seedlings germinating on sucrose-deficient media [35]. In this study, we found that the deletion of *FfATPh*, *FfATP5*, and *FfATPb* also severely affected the nutritional growth of *F. fujikuroi*. The vegetative growth rate of *∆FfATPh*, *∆FfATP5*, and *∆FfATPb* significantly decreased on PDA, CM, V8, and MM medium. Meanwhile, the sporulation ability of the deletion mutants also significantly decreased. Taken together, *FfATPh*, *FfATP5*, and *FfATPb* are essential for vegetative growth and asexual reproduction.

The typical symptom of RBD is that the diseased seedlings are higher than the healthy seedlings [36]. In addition, the disease seedlings are sterile and cannot produce edible grains. The plant hormone (GA) produced by *F. fujikuroi* will continuously stimulate the growth of rice plants and contribute to the production of bacterial toxins [2,37]. In *FfATPh*, *FfATP5*, and *FfATPb* deletion mutants, the structure of ATP synthase is disrupted, leading to impaired energy synthesis and the reduced pathogenicity to *F. fujikuroi*. However, further research is needed to determine whether the decrease in pathogenicity will result in a reduction in gibberellin synthesis in *F. fujikuroi*.

In addition, *FfATPh*, *FfATP5*, and *FfATPb* nonspecifically regulated the sensitivity of *F. fujikuroi* to different fungicides. The sensitivity of *∆FfATPh*, *∆FfATP5*, and *∆FfATPb* to fludioxonil, phenamacril, pyraclostrobine, and fluazinam decreased. We speculate that the reason for the decrease in fungicide sensitivity is the destruction of ATP synthase structure. Moreover, *FfATPh*, *FfATP5*, and *FfATPb* are involved in the regulation of the sensitivity to various stresses. In *Saccharomyces cerevisiae*, the lack of *ATPh* affects the assembly of ATP6 into ATP synthase, leading to membrane potential dispersion. In this study, the sensitivity of *ΔFfATPh* to ionic stress and osmotic stress decreased significantly. Research has found that ATP synthase, one of the mitochondrial proteins sensitive to H_2_O_2_, plays a crucial role in oxidative stress. When the mitochondrial ATP synthase inhibitor oligomycin reduces intracellular free ATP levels, it can reduce the viability of fungi under oxidative stress [38]. In *F. fujikuroi*, *ΔFfATPb* and *ΔFfATP5* exhibit more sensitivity to oxidative stress. We speculated that the absence of *FfATPb* and *FfATP5* may lead to the dysfunction of ATP synthase in the fungal body, making it more sensitive to oxidative stress.

In this study, *FfATPh*, *FfATP5*, and *FfATPb* are located in mitochondria and they interacted to form the peripheral stalk of ATP synthase. Experimental evidence has demonstrated that the deletion of these three genes leads to alterations in mitochondrial localization in *F. fujikuroi*. In *Saccharomyces cerevisiae*, when the entire N-terminal domain of ATPb is missing, mitochondrial subunit g is almost non-existent. The mutated ATP synthase does not dimerize or oligomerize, and the mutated cells exhibit an abnormal onion-like mitochondrial morphology [39]. In conclusion, *FfATPh*, *FfATP5*, and *FfATPb* are crucial for the localization of mitochondria. Currently, we have only observed the structure of mitochondria. Next, we need to further investigate whether gene deletions will affect the internal morphology of mitochondria, such as cristae.

In summary, *FfATPh*, *FfATPb*, and *FfATP5* are involved in regulating development, asexual reproduction, and pathogenicity, which are essential for the function of *F. fujikuroi*. At the same time, the mounting evidence emphasizes ATP synthase as a key molecule and enzyme catalyst in the balance between cell survival and death. This highlights the increasing attractiveness of ATP synthase as a pharmacological target for fungicide development. By manipulating these three subunits, we aimed to understand the role of ATP synthase in energy production and its impact on the growth and survival of fungi. Therefore, in the subsequent experiments, further research is needed to study the specific impact of the deletion of these three subunits on the ATP synthesis level and the functionality of other subunits of ATP synthase. Only through these investigations can we enhance our study and provide valuable insights into the potential of ATP synthase as a target for controlling fungal growth.

## 4. Materials and Methods

### 4.1. Strains, Media, and Fungicides

The *F. fujikuroi* wild-type strain A was preserved in the laboratory, and the deletion mutants were obtained from strain A. The culture media involved in this study included potato dextrose agar (PDA), minimal medium (MM), complete medium (CM), V8 juice medium (V8), CMC liquid medium, and YEPD medium [40].

Tebuconazole (96%, Ningbo Sanjiang Yinong Chemical Co., Ltd., Ningbo, China), fludioxonil (95%, Swiss Syngenta crop Protection Co., Ltd., Basel, Switzerland), prochloraz (95%, Nanjing Red Sun Co., Ltd., Nanjing, China), phenamacril (95%, Jiangsu Pesticide Research Institute Co., Ltd., Nanjing, China), pyraclostrobine (98%, Anhui Guangxin Agrochemical Co., Ltd., Anhui, China), fluazinam (98%, Nanjing Red Sun Co., Ltd., Nanjing, China) were used in this experiment.

### 4.2. Construction of Deletion Mutant Vectors, GFP, and Flag Fusion Cassettes

The deletion mutants of *FfATPh*, *FfATP5*, and *FfATPb* were constructed by using homologous recombination. The upstream fragment of *FfATPh*, about 1 kb, and the downstream fragment of *FfATPh*, about 1 kb, were amplified from the genome DNA of WT with primers listed in Appendix A. The 3490 bp HPH-HSV-tk cassette, which contains the hygromycin-resistance gene, the herpes simplex virus thymidine kinase gene, and the Aspergillus nidulans trpC promoter was amplified from the hph-hsv plasmid DNA with primers SS-F/SS-R. Then, the upstream and downstream fragments and HPH-Hsv-tk fragment were fused using La Taq Polymerase (Baori Medical Biotechnology Co., Ltd., Beijing, China) without primers [41]. The fusion vector was used for transformation into *F. fujikuroi* wild-type strain by using protoplast transformation as previously reported [42]. The transformants were identified via PCR assays and further verified through Southern blotting (Appendix A). *∆FfATPb* and *∆FfATP5* strains were also obtained through the aforementioned experimental methods in this study. All the primers used are shown in Appendix A.

To construct GFP and Flag fusion cassettes, the *FfATPh*, *FfATP5*, and *FfATPb* fragments containing the native promoter and ORF (without the stop codon) were amplified with the primers FfATPh (FfATPb, FfATP5)-RP-F/FfATPh (FfATPb, FfATP5)-RP-F, and then, the resulting PCR products were restructured into XhoI-digested pDL2 using a 2xMultiF Seamless Assembly Mix (ABclonal Technology Co., Ltd., Wuhan, China). Subsequently, the completed plasmid was transferred into *Escherichia coli* DH5α for amplification, with Escherichia coli transformation performed as previously reported [43]. And then, the FfATPh (FfATPb, FfATP5)-GFP fusion vector was added to the protoplast of the wild-type strain to obtain the FfATPh (FfATPb, FfATP5)-GFP strain. The green fluorescent signal was taken under a confocal microscope (Leica TCS SP8, Germany).

### 4.3. Vegetative Growth and Asexual Reproduction Assays

To determine the growth of each strain on different media, the wild-type strain and the deletion mutant strains were cultured on PDA, CM, MM, and V8 and incubated at 25 °C for 7 days. The experiment was repeated three times independently.

For the asexual reproduction assay, three 5 mm plates of each strain were transferred to 100 mL of CMC liquid medium and cultivated under light at 25 °C, with 175 rpm, for five days. The number of spores of each strain was determined using a hemocytometer. The experiment repeated independently three times.

### 4.4. Sensitivity of Strains to Different Fungicides and Stress Factors

To determine the sensitivity to different fungicides, we inoculated each strain on PDA plates containing 0.125 μg/mL tebuconazole, 0.0625 μg/mL fludioxonil, 0.1 μg/mL prochloraz, 0.25 μg/mL phenamacril, 5 μg/mL pyraclostrobine, and 0.25 μg/mL fluazinam. After cultivation at 25 °C for 7 days in the dark, and then measured the colony diameter and calculated the growth inhibition rate at the concentration of the fungicide. Each experiment was repeated three times.

For stress sensitivity assays, we inoculated each strain on PDA plates containing 0.7 M NaCl, 0.7 M KCl, 5 mM ZnCl_2_, 0.5 M CaCl_2_, 0.5 M MgCl_2_, 1.2 M Sor, 0.03% SDS, 0.1 mM menadione, and 500 μg/mL Congo Red. After incubating at 25 °C in the dark for 7 days, we measured the colony diameter and took photos. The experiment was repeated three times independently.

The hyphal growth inhibition rate was calculated using the following formula: (diameter of control group diameter of treatment group)/(diameter of control group diameter of the culture dish) × 100%.

### 4.5. Pathogenicity Assay

Pathogenicity assays on rice were conducted as previously described [40]. The mycelial plugs were inoculated in test tubes, and each sample was repeated 10 times. After being cultured in an incubator at 28 °C, with 12 h of light and 12 h of darkness, for 7 days, the length of the sprout sheath of rice seedlings was measured and photos were taken. The experiment was repeated three times.

### 4.6. Microscopic Examinations

The hyphal morphology was observed with an inverted fluorescent microscope. The localization of each strain in the hyphae and conidia was observed with a TCS SP8 confocal microscope (Leica TCS SP8, Germany). The strain was cultured in the liquid media YEPD for 36 h, and the conidia were cultured in the liquid media CMC for 4 d.

To observe the localization of mitochondria in wild-type A and gene deletion mutants, all strains were cultured in a YEPD medium for 72 h. Then, the mycelia were stained with Mitro-Tracker and observed with a TCS SP8 confocal microscope (Leica).

### 4.7. Western Blotting Hybridization

The preparation of protein samples was performed using previously reported methods [44]. For about 80 μL protein samples, a quarter volume of 5× loading buffer was added, and then, it was boiled for ten minutes. Subsequently, 15 μL of the above samples was taken and loaded onto 10% SDS-PAGE gels and transferred to an Immobilon-P transfer membrane (Millipore, USA). The monoclonal anti-GFP antibody 300943 (Zenbio, Shanghai, China) and monoclonal anti-Flag antibody 390002 (Zenbio, Shanghai, China) were used at a 1:1000 dilution for immunoblot analyses. Samples were incubated with the corresponding secondary antibodies, and exposure was performed with the chemiluminescence method.

### 4.8. Bimolecular Fluorescent Complimentary (BiFC) Assay

The correctly sequenced plasmids with FfATPh (FfATPb, FfATP5)-pHZ65 and FfATPb (FfATP5, FfATPh)-pHZ68 were co-transformed into the parental strain A. After that, the obtained transformants were screened as resistant to bleomycin and hygromycin, and the transformants that could grow on the resistance plate were selected. Then, whether there was green fluorescence in the mycelium was observed under a fluorescence confocal microscope. If there is green fluorescence, this indicates an interactive relationship between the two genes. In this experiment, the pHZ65 and pHZ68 plasmids, introduced separately, were used as controls, and the control strains did not exhibit green fluorescence.

### 4.9. Co-Immunoprecipitation (Co-IP) Assay

The GFP and 3× Flag fusion constructs were verified via DNA sequencing and co-transformed into A. The expression level of fusion constructs was determined via Western blotting hybridization. In addition, the transformants expressing a single fusion construct were used as references. For Co-IP assays, magnetic beads were first incubated with the anti-GFP antibody for 20 min. Afterward, the magnetic beads were incubated with total protein samples. Proteins eluted from agarose were assayed via Western blotting with anti-GFP or anti-Flag antibodies. All experiments were repeated twice.

### 4.10. Yeast Two-Hybrid Assay (Y2H)

To construct plasmids for Y2H analyses, the coding sequence of each gene was amplified from the cDNA of A and inserted into the yeast GAL4-binding domain vector pGBKT7 and GAL4 activation domain vector pGADT7 (Clontech, USA). The pairs of Y2H plasmids were co-transformed into *S. cerevisiae* AH109 following the LiAc/SSDNA/PEG transformation protocol. Besides, the plasmid pairs pGBKT7-Lam and pGADT7 served as a negative control, and plasmid pairs pGBKT7-53 and pGADT7 were used as a positive control. Transformants were grown on synthetic dropout medium (SD) lacking Leu and Trp at 30 °C for 3 days, and then transferred to SD without His, Leu, Trp, and Ade. Other vectors were obtained using similar methods. The experiment was independently repeated three times.

## Figures and Tables

**Figure 1 ijms-24-13273-f001:**
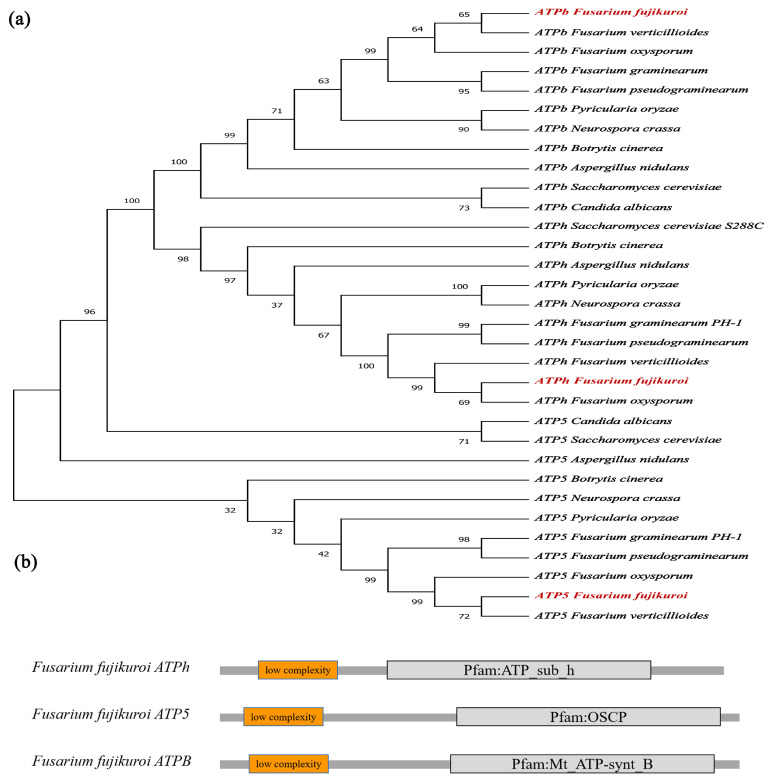
Identification of *FfATPh*, *FfATP5*, and *FfATPb* in *F. fujikuroi*. (**a**) Phylogenetic analysis of *FfATPh*, *FfATP5*, and *FfATPb* from *Fusarium fujikuroi*, *Fusarium oxysporum*, *Fusarium verticillioides*, *Fusarium pseudograminearum*, *Fusarium graminearum*, *Neurospora crassa*, *Pyricularia oryzae*, *Saccharomyces cerevisiae*, *Aspergillus nidulans*, and *Botrytis cinerea*. Amino acid sequences were obtained from NCBI, and MEGA-X was used to construct the phylogenetic tree. The *FfATPh*, *FfATP5*, and *FfATPb* from *Fusarium fujikuroi* are highlighted in red. (**b**) The domains of FfATPh, FfATPb, and FfATP5 in *F. fujikuroi*. A simple modular architecture research tool (SMART) database was used to find the domains.

**Figure 2 ijms-24-13273-f002:**
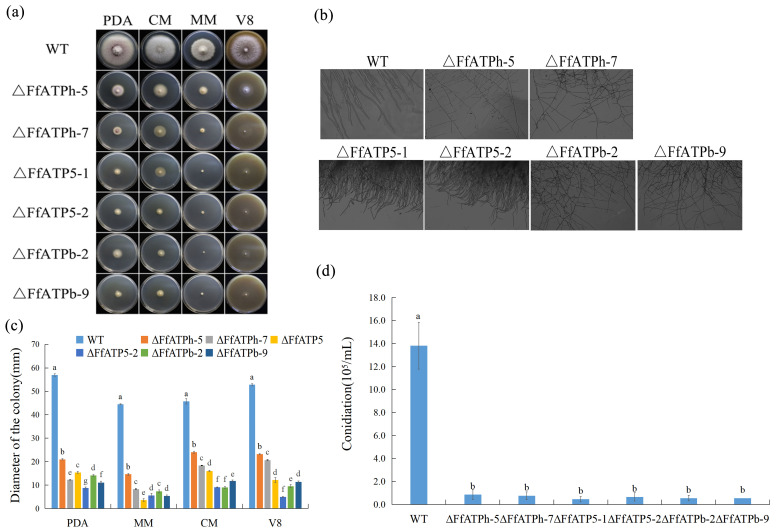
Vegetative growth of *FfATPh*, *FfATP5*, and *FfATPb* deletion mutants. (**a**) The colonial morphology of strains cultured on PDA, CM, MM, and V8 medium at 25 °C for 5 days. (**b**) The inverted microscope (20×) was used to observe the mycelium tip morphology of strains cultured for 3 days in YEPD liquid medium. (**c**) The hyphal growth rate of the strains on PDA, CM, MM, V8, and YBA medium. Error bars denote the standard error of three repetitions. LSD method for significant difference analysis; bars with the same letter indicate no significant difference at *p* = 0.05. (**d**) The sporulation quantity of A, *FfATPh*, *FfATP5*, or *FfATPb* deletion mutants. Error bars denote the standard error of three repetitions. LSD method for significant difference analysis; bars with the same letter indicate no significant difference at *p* = 0.05.

**Figure 3 ijms-24-13273-f003:**
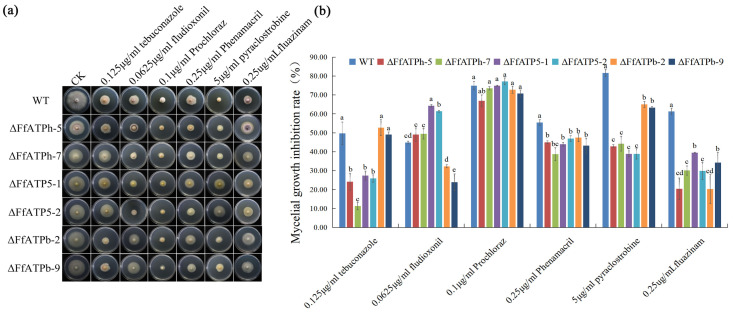
The sensitivity of *ΔFfATPh*, *ΔFfATP5*, and *ΔFfATPb* to different fungicides compared with the WT. (**a**) Effects of 0.125 μg/mL tebuconazole, 0.0625 μg/mL fludioxonil, 0.1 μg/mL prochloraz, 0.25 μg/mL phenamacril, 5 μg/mL pyraclostrobine, and 0.25 μg/mL fluazinam on mycelial linear growth of *ΔFfATPh*, *ΔFfATP5*, and *ΔFfATPb* mutants. (**b**) The hyphal growth rate of the strains on PDA supplemented with each fungicide. Error bars denote the standard error of three repetitions. LSD method for significant difference analysis; bars with the same letter indicate no significant difference at *p* = 0.05.

**Figure 4 ijms-24-13273-f004:**
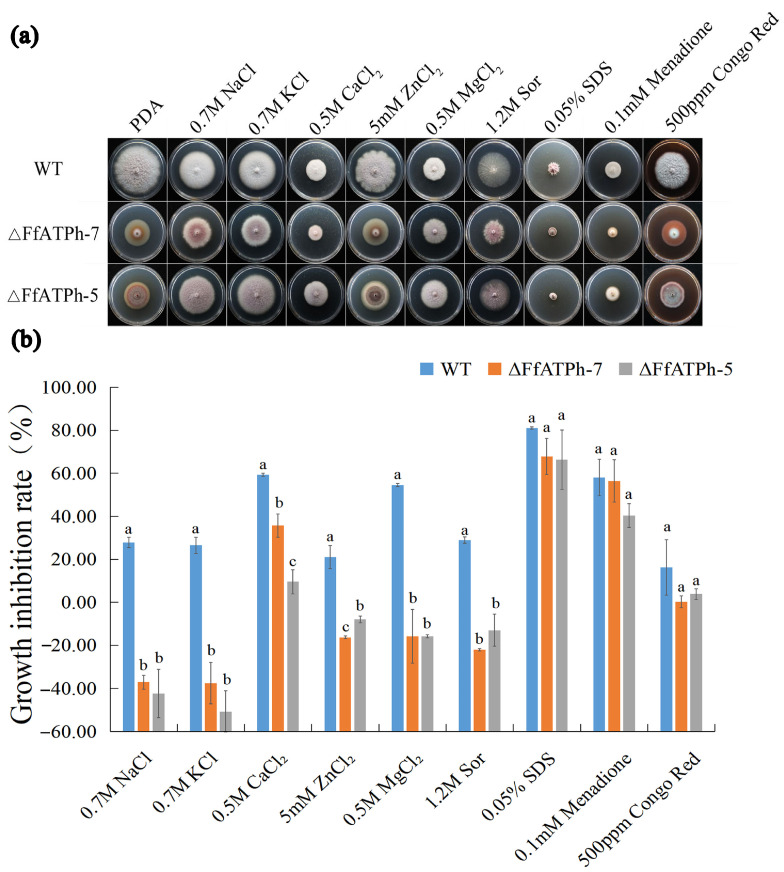
Response of *ΔFfATPh* to different stress factors. (**a**) The colonial morphology of different strains on PDA medium with 0.7 M NaCl, 0.7 M KCl, 0.5 M CaCl_2_, 5 mM ZnCl_2_, 0.5 M MgCl_2_, 1.2 M sorbitol, 0.03% SDS, 0.1 mM menadione, and 500 µg/mL Congo red for 7 days. (**b**) Hyphal growth rate of the strains on different stress factor media for 7 days. Error bars denote the standard error of three repetitions. LSD method for significant difference analysis; bars with the same letter indicate no significant difference at *p* = 0.05.

**Figure 5 ijms-24-13273-f005:**
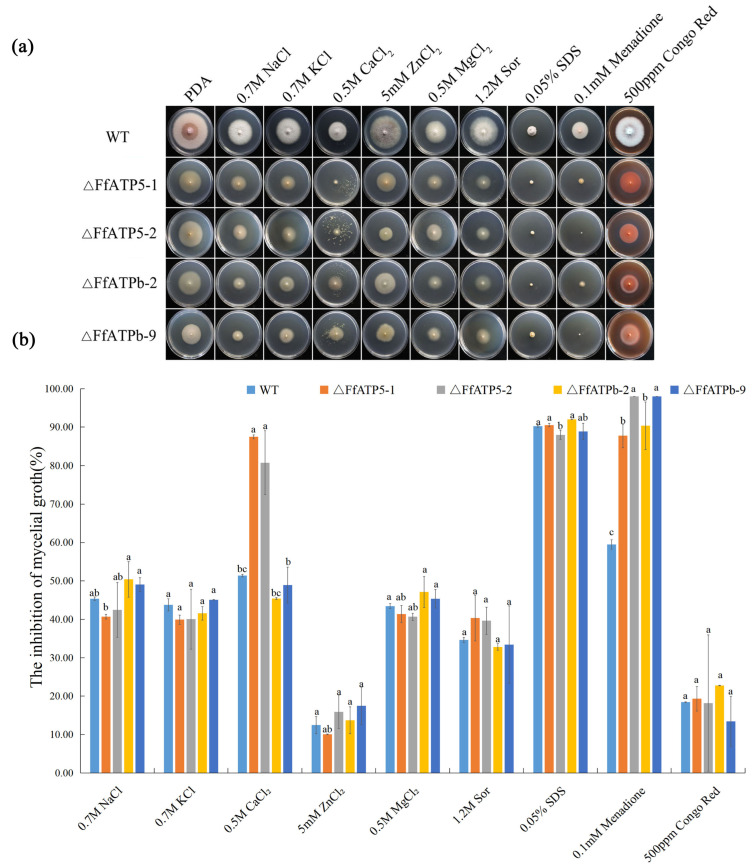
Response of *ΔFfATP5* and *ΔFfATPb* to different stress factors. (**a**) The colonial morphology of different strains on PDA medium with 0.7 M NaCl, 0.7 M KCl, 0.5 M CaCl_2_, 5 mM ZnCl_2_, 0.5 M MgCl_2_, 1.2 M sorbitol, 0.03% SDS, 0.1 mM menadione, and 500 µg/mL Congo red for 7 days. (**b**) Hyphal growth rate of the strains on different stress factor media for 7 days. Error bars denote the standard error of three repetitions. LSD method for significant difference analysis; bars with the same letter indicate no significant difference at *p* = 0.05.

**Figure 6 ijms-24-13273-f006:**
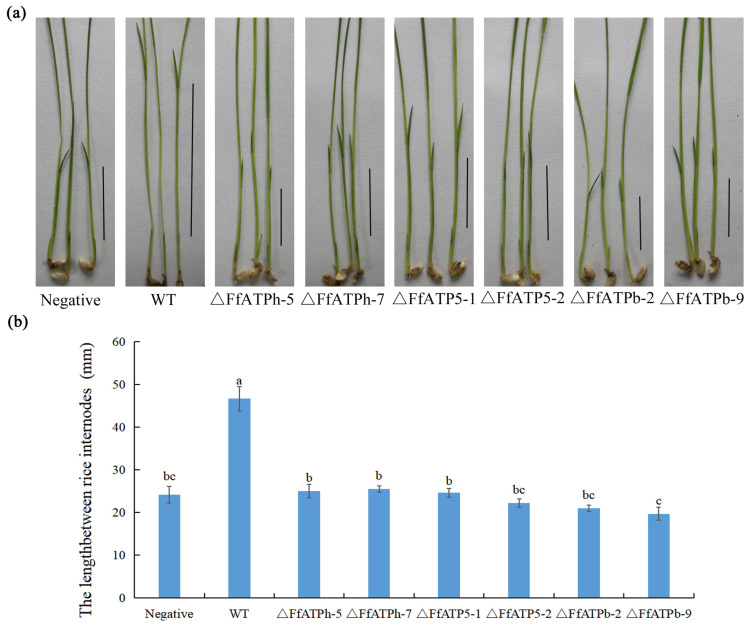
*FfATPh*, *FfATPb*, and *FfATP5* are indispensable for pathogenicity. (**a**) All strains were inoculated on the root of rice seedlings, and the length of the rice seedling sheath was measured 7 days later. A was a positive control and PDA was the negative control. (**b**) The length of the rice seedling sheath after inoculation with different strains. Error bars denote the standard error of three repetitions. LSD method for significant difference analysis; bars with the same letter indicate no significant difference at *p* = 0.05.

**Figure 7 ijms-24-13273-f007:**
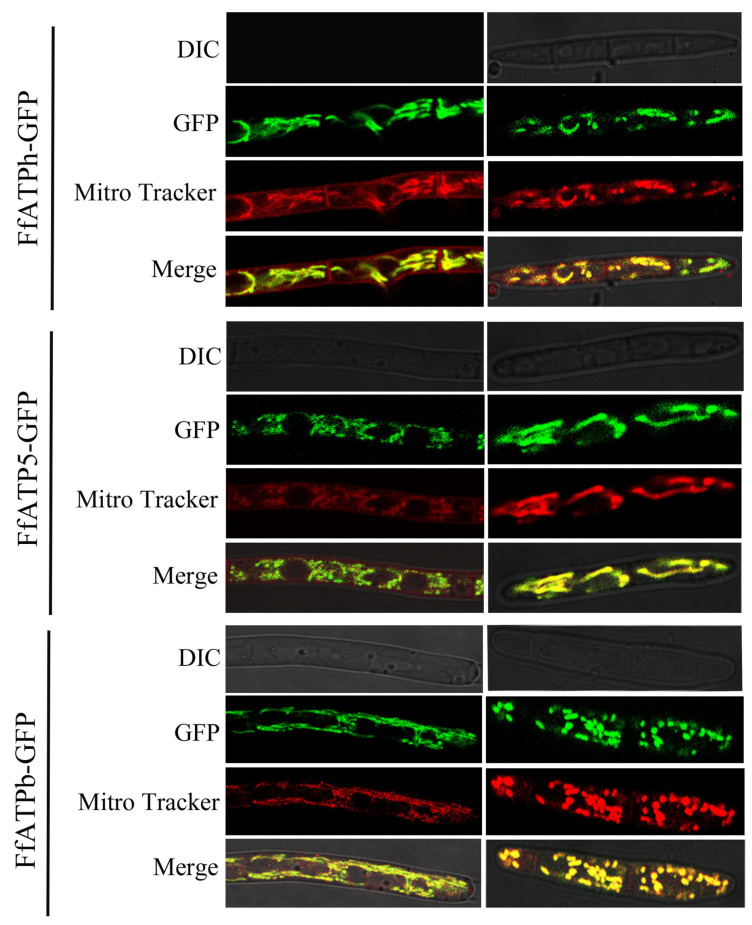
*FfATPh*, *FfATP5*, and *FfATPb* are located in the mitochondria. The strains were incubated in YEPD for 36 h and CMC for 5 d to obtain the mycelium and conidia. The image was captured under a TCS SP8 confocal microscope at 100× magnification. *FfATPh* (*FfATP5*, *FfATPb*)—GFP—green fluorescence, *FfATPh* (*FfATP5*, *FfATPb*)—Mitro-Tracker (Mitochondrial dye)—red fluorescence, *FfATPh* (*FfATP5*, *FfATPb*)—Merge—yellow fluorescence.

**Figure 8 ijms-24-13273-f008:**
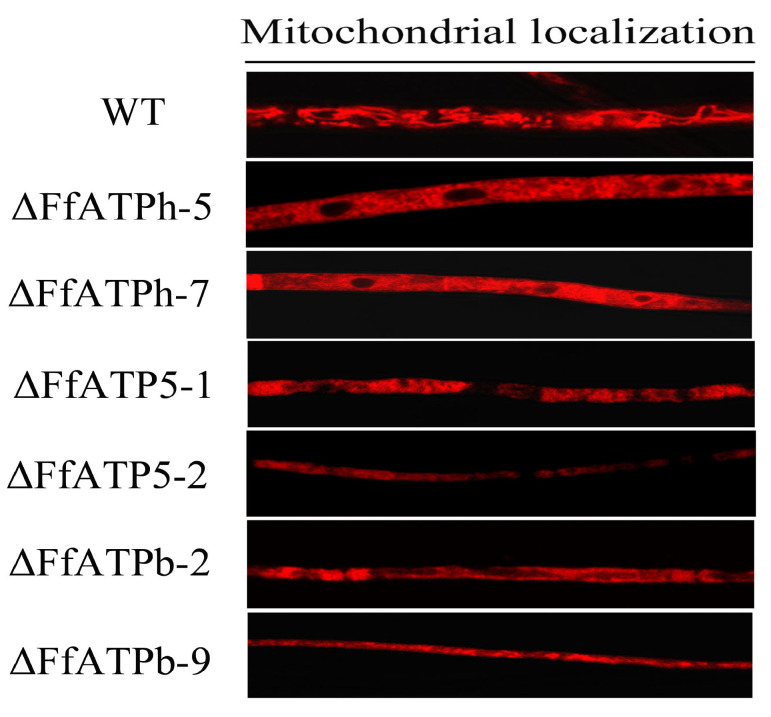
*FfATPh*, *FfATP5*, and *FfATPb* affect the localization of mitochondria. The mycelia of different strains were collected after being cultured in YEPD liquid medium for 36 h. Mitochondria were stained with Mito-tracker (Red fluorescence), and then, the fluorescence was observed with a Leica-TCS-SP8 confocal microscope (100×).

**Figure 9 ijms-24-13273-f009:**
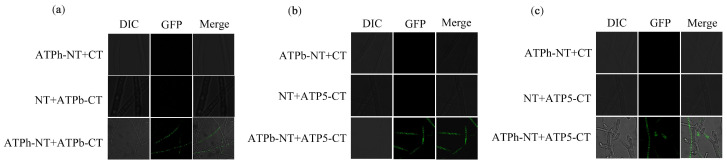
Interaction among FfATPh, FfATP5, and FfATPb. (**a**) The interaction of FfATPh and FfATPb was confirmed via bimolecular fluorescence complementation (BiFC) analysis. The image was captured under a TCS SP8 confocal microscope at 100× magnification. The strain of ATPh-NT+ATPb-CT displays green fluorescence. (**b**) BiFC confirmed the interaction of FfATP5 and FfATPb. The image was captured under a TCS SP8 confocal microscope at 100× magnification. The strain of ATPb-NT+ATP5-CT displays green fluorescence. (**c**) BiFC confirmed the interaction of FfATPh and FfATP5. The image was captured under a TCS SP8 confocal microscope at 100× magnification. The strain of ATPh-NT+ATP5-CT displays green fluorescence.

**Figure 10 ijms-24-13273-f010:**
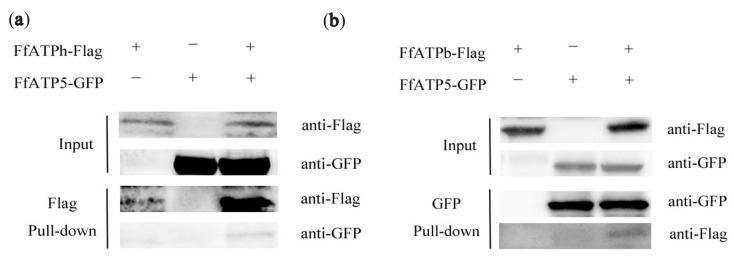
Interaction among *FfATPh*, *FfATP5*, and *FfATPb*. (**a**) Co-immunoprecipitation confirmed the interaction between FfATPb and FfATP5. (**b**) Co-immunoprecipitation confirmed the interaction between FfATPh and FfATP5.

## Data Availability

The data that support the findings of this study are available from the corresponding author upon reasonable request.

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
