# Peer review of "The ATP Synthase Subunits FfATPh, FfATP5, and FfATPb Regulate the Development, Pathogenicity, and Fungicide Sensitivity of Fusarium fujikuroi"

_ijms, 2023, doi:10.3390/ijms241713273_

Round 1

Reviewer 1 Report

Yang et al detail a study of three ATP synthase genes in Fusarium fujikuroi by using knockout mutants as well as protein related methods to study the interaction of all three genes in the fungus. While plenty of information about how these proteins interact as well as the phenotypes, they govern in F. fujikuroi most of this work is based on the creation of multiple mutant strains or tagging genes. Currently no information is presented either as a main figure, in the supplemental, or even in the materials to confirm that these strains are correct. This is issues is compounded by the lack of complement strains for the deletion mutant. This information/data is required for this manuscript and needed to validate the work. Additionally, if no complement is made multiple mutants need to be verified and shown to have the same phenotype.

Other Comments:

In introduction author mentions three units of the F1F0-ATP synthase, it would be useful to take the F. fujikuroi ATPases and determine which of these correspond to each of the three units. This would be beneficial in the discussion when comparing the observed phenotypes to other studies.

The introduction also introduces other ATPases studied in other fungal species, for Figure 1 it would be more informative to include these genes or denote their inclusion if already present since the names from other species don’t match your labels.

Figure 3b some of the letters used to denote statistical significance are obscured by the error bars. This also occurs in Figure 4b but additionally here the x-axis labels are obscured with the data bars.

In multiple figures a label of “A” is used without any contextual meaning. It could be explained in the legends, but I think it would be better to change label to control or WT.

In the pathogenicity assay, PDA is used to describe the negative control, based on the methods I assume this to be just a plug of PDA with no fungal growth. If correct, please try and make this a bit clearer of the meaning. The previous change of label A may also be sufficient to clarify this as well.

Reviewer 2 Report

Fear authors

Happy day

This study concern with ATP synthase subunits FfATPh, FfATP5, and FfATPb regulate 2 the development, pathogenicity, and fungicide sensitivity of Fusarium fujikuroi. The authors conduct a study to characterize the subunits of the peripheral stalk FfATPh, FfATP5, and FfATPb and explored their functions on development and path-ogenicity of Fusarium Fujikuroi. The FfATPh, FfATP5, and FfATPb deletion mutations (ΔFfATPh, ΔFfATP5, and ΔFfATPb) presents deficiencies in vegetative growth, sporulation, and pathogenicity. This study is helpful to understand the mechanism of fungal energy syn-259 thesis and provide new ideas for the study of new preparations.

The paper needs some improvement

1-      The images are not clear due to small space given to it.

2-      Light microscope images are not clear and bigger magnifications are needed to show any differences. The differences in the colonial morphology of strains are nit clear.

3-      In line 284 the authors use the future “ we will amplify.”

a.       The authors need to show that they have already made this experiment in this study and did not use any conversion words that might reflect that part of this work has been done before. For example, in the end of line 280 it is better to say ‘’in this study’’.

b.      Data from lines 280 to 288 need more space and more details. That because it includes the main technique to induce mutations.

4-      More details are needed in the discussion part to link between each single conducted experiment and what could such experiments tell us concerning to their results and their relationship with the main aim of this study.

5-      Two points should add to the discussions

a.       What could do in future studies and how our study could benefit the farmers or the rice producer. What you are recommended based on this study.

b.      What points which are not clear and need additional experiments in future studies.

In general, this paper should be accepted but it could be improved without conducting any further experiments.

Round 2

Reviewer 1 Report

Authors have addressed all my concerns and in my opinion needs no further changes.